# miR-455-3p has superior diagnostic potential to PSA in peripheral blood for prostate cancer

Yi Cen[1,2☯], Shourui Feng[3☯], Yuyu Xu[1], Churuo Zhang[1], Xiangjin Lin[1], Xuan Ye[4], Zeyu Zha[5], Haiyan Wang[6]*, Guangbin Zhu[1]*

1 Department of Medical Imaging, The Fifth Affiliated Hospital of Guangzhou Medical University, Guangzhou, P. R. China, 2 Guangdong Provincial Key Laboratory of Molecular Target & Clinical Pharmacology, the NMPA and State Key Laboratory of Respiratory Disease, Guangzhou Medical University, Guangzhou, P. R. China, 3 State Key Laboratory of Biocontrol, School of Life Sciences, Sun Yat-sen University, Guangzhou, P. R. China, 4 Department of Thyroid and Breast Surgery, Guangzhou Women and Children's Medical Center, Guangzhou Medical University, Guangzhou, P. R. China, 5 Department of Urology, Second Affiliated Hospital of Bengbu Medical College, Bengbu, P. R. China, 6 Shenzhen Bao'an Chinese Medicine Hospital, Guangzhou University of Chinese Medicine, Shenzhen, P. R. China

☯ These authors contributed equally to this work.
* guangbin_zhu@163.com (GZ); why6838260@163.com (HW)

**Data Availability Statement:** All relevant data are within the manuscript and its Supporting Information files.

**Funding:** National Natural Science Foundation of China (Grant Number: 82201522) Funder: Xuan Ye

## Abstract

### Background

Prostate-specific antigen (PSA) is commonly used as a biomarker to diagnose and predict the course of prostate cancer (PCa). However, PSA detection is susceptible to changes in the physiologic environment, which may lead to some misdiagnosis. Thus, it is crucial to find a novel diagnostic marker.

### Methods

We accessed microRNA (miRNA) expression datasets (GSE206793 and GSE112264) from the GEO database, analyzing peripheral blood samples from PCa patients. Differentially expressed miRNAs (DEmiRNAs) were identified using GEO2R. A specific miRNA, miR-455-3p, was pinpointed through rigorous analysis of clinical correlations and ROC curves. Peripheral blood samples from healthy individuals and PCa patients were subjected to qRT-PCR validation, aligning results with the GSE206793 dataset. The miRWalk database was utilized to predict downstream genes, while STRING facilitated the construction of a protein-protein interaction (PPI) network. KEGG pathway analysis enriched our understanding of potential molecular pathways.

### Results

We found that miR-455-3p was highly expressed in the peripheral blood of PCa patients with Gleason score (GS) $\geq$ 8, while independent of T stage, age and PSA. ROC analysis revealed a favorable diagnostic efficacy of miR-455-3p and AUC for the two datasets was respectively 0.943 and 0.847. The qRT-PCR assay also revealed consistent results. Interestingly, the PSA levels of P1 (GS = 5 + 4) and P6 (GS = 3 + 3) were respectively 3.38 and 4.45 ng/ml, while miR-455-3p was highly expressed in both, suggesting its low misdiagnosis. The speculation was validated in GSE206793 dataset. Finally, 9 potential targets of

Funder' Role: the acquisition of data. Plan on enhancing scientific research in GMU Funder: Guangbin Zhu Funder' Role: conceived and designed the study.

**Competing interests:** NO authors have competing interests Enter: The authors have declared that no competing interests exist.

**Abbreviations:** PSA, Prostate-specific antigen; PCa, Prostate cancer; miRNAs, microRNAs; GEO, Gene Expression Omnibus; DEmiRNAs, Differentially expressed miRNAs; GS, Gleason score; T, Tumor; ROC, Receiver operating characteristic; AUC, Area under the curve; qRT-PCR, quantitative reverse transcriptase PCR; PPI, Protein-protein interaction; KEGG, Kyoto Encyclopedia of Genes and Genomes; UTR, Untranslated region; mRNAs, messenger RNAs; FC, Fold change; MRI, Magnetic resonance imaging; FDR, False discovery rate; SD, Standard deviation; LC, Liver cancer; T2D, Type 2 diabetes; PPP2R2A, Protein phosphatase 2 regulatory subunit Balpha; ITGB1, Integrin subunit beta 1; CDKN1A, Cyclin dependent kinase inhibitor 1A.

miR-455-3p were predicted. PPI network revealed PPP2R2A, ITGB1 and CDKN1A as key nodes. KEGG pathway analysis revealed that they were enriched in various cancers, biological processes and molecular signals.

## Conclusion

Our study identifies miR-455-3p as a promising diagnostic marker for PCa, outperforming PSA in terms of specificity and sensitivity. The robustness of miR-455-3p, coupled with its potential downstream targets and associated pathways, highlights its clinical significance for improved PCa diagnosis and management.

## Introduction

Prostate cancer (PCa) is one of the common malignant tumors in the male urinary system worldwide, whose incidence and mortality are increasing year by year [1]. What's worse is that the bone metastasis and recurrence commonly occur, which makes the prognosis poor [2]. The basic diagnosis of PCa includes rectal digital examination detection, serum prostate-specific antigen (PSA) detection, biopsy analysis and histological analysis [3]. However, the PCa progress and unnormal hyperplasia are difficult to be verified by these methods [4, 5]. In particular, PSA results are susceptible to drugs, inflammation and benign prostatic lesions, leading to a lack of specificity and sensitivity in PCa prognosis [6]. Therefore, it is crucial to find a novel clinical diagnostic marker.

microRNAs (miRNAs) are the single-stranded non-coding RNAs with high conservative. They were constructed with a length of about 18 to 22 nucleotides [7]. miRNAs interfere with protein translation by binding to the sequences in the 3' untranslated region (UTR) of messenger RNAs (mRNAs), reducing the stability of mRNAs or inhibiting the expression of the transcription genes [8]. Participating in various physiological processes such as cell proliferation and apoptosis, it plays an important regulatory role in diseases and is closely related to the occurrence of various tumors [9–11]. Circulating miRNAs have received increasing attention in recent years as diagnostic markers for various diseases due to their convenience in monitoring [12–14]. Studies have shown that circulating miRNAs are complementary candidate biomarkers for PCa diagnosis.

With the development of sequencing technology, bioinformatics has been used to explore the pathological mechanisms of various diseases at the gene level. Gene Expression Omnibus (GEO) database is an online gene chip database of gene expression in species [15]. Gene maps and microarrays are used to screen for differentially expressed miRNAs (DEmiRNAs) and genes. However, the research on biomarkers is still insufficient. The present study identified a common target, miR-455-3p, via interaction analysis of two GEO datasets. miR-455-3p was then analyzed for correlation with clinical features in PCa and validated with clinical samples. The downstream binding genes of miR-455-3p were further predicted. Finally, a protein-protein interaction (PPI) network was constructed for the target genes and the Kyoto Encyclopedia of Genes and Genomes (KEGG) pathway analysis was performed.

## Materials and methods

### Data collection

GSE206793, GSE112264 and GSE138740 datasets from the GEO database (ncbi.nlm.nih.gov/geo/) were selected for subsequent analysis. Samples for GSE206793 and GSE112264 datasets

were obtained from peripheral blood of healthy individuals and cancer patients. GSE206793 dataset included 5 healthy individuals, 40 low-risk, 16 medium-risk and 91 high-risk PCa patients with clinical informations (including age, Gleason score (GS) and PSA). GSE112264 dataset included 41 healthy individuals, 809 PCa patients with different tumor (T) stages (T1 to T4) and 50 each of 10 other cancer patients (including biliary tract cancer, bladder cancer, colorectal cancer, esophageal cancer, gastric cancer, glioma, hepatocellular carcinoma, lung cancer, pancreatic cancer and sarcoma). GSE138740 dataset included total RNA sequencing profiles of urinary exosomes from 89 normal individuals and 146 PCa patients with GS informations. The three datasets above were built on GPL19117 (Affymetix Multispecies miRNA-4 Array), GPL21263 (3D-Gene Human miRNA V21_1.0.0) and GPL21572 (Affymetrix Multispecies miRNA-4 Array) platforms, respectively.

## DEmiRNAs screening

GEO2R is an interactive network tool, which can compare and analyze multiple samples under the same experimental conditions. Gene expression matrices were analyzed and downloaded online using GEO2R. For GSE206793 dataset, $|\log2 (\text{fold change, FC})| > 1$ and $P < 0.05$ were used as the screening conditions. For the GSE112264 dataset, which had a large screening range due to its large cases, $|\log2 (\text{FC})| > 3$ and $P < 0.01$ were used as screening conditions for dimensionality reduction. The DEmiRNAs screened from the two datasets were interacted to screen for a target miRNA (miR-455-3p).

## Clinical feature analysis

In GSE206793 dataset, miR-455-3p expression was analyzed in PCa patients with different risk degrees. PCa patients were further divided into two groups based on age ($< 60$ vs $\geq 60$), GS ($< 8$ vs $\geq 8$) and PSA ($< 4$ vs $\geq 4$) for comparison. In GSE112264 dataset, miR-455-3p expression was analyzed in PCa patients with different T stages and compared with other cancer patients. The receptor operating characteristic (ROC) of miR-455-3p in both datasets was further analyzed using the pROC R package to evaluate its diagnostic efficacy.

## Correlation analysis

In GSE206793 dataset, correlation between PSA level and miR-455-3p expression was examined using the Pearson's method. Due to the large range of PSA concentration, it was represented as log (PSA concentration). After excluding samples with no PSA information and PSA concentration of 0 from the 147 samples, the remaining 134 samples were used for correlation analysis.

## Clinical sample collection

Peripheral blood samples were collected from 8 prostate cancer (PCa) patients diagnosed at the Department of Urology, Fifth Hospital of Guangzhou Medical University, and from 4 healthy individuals between January 3, 2024, and February 23, 2024. Magnetic resonance imaging (MRI) films and pathological information for the PCa patients were provided by the Department of Radiology, Fifth Affiliated Hospital of Guangzhou Medical University. The study was approved by the Ethics Committee of the Fifth Hospital of Guangzhou Medical University (approval number: GYWY-K2024-04).

## Quantitative reverse transcription PCR (qRT-PCR) assay

Total RNA was extracted from peripheral blood samples obtained from 8 prostate cancer (PCa) patients diagnosed at the Department of Urology, Fifth Hospital of Guangzhou Medical

University, and from 4 healthy individuals, using Trizol reagent (BioTeke, China). After determining the concentration and purity of RNA samples, RNA reverse transcription was performed according to the steps of reverse transcription kit (Vazyme, China). The synthetic cDNAs were used as template for fluorescence detection using SYBR Green qPCR detection kit (Biosharp, China). The results were quantified using the relative quantitative $2^{-\Delta\Delta CT}$ method [16]. U6 as an internal control gene and the relative miR-455-3p expression was normalized to U6 expression. Primer sequences of miR-455-3p: $5'$-ACACTCCA GCTGCAGTCC ATGGGCAT-$3'$ (F) and $5'$-ACTGGTGTCGTGGAGTCGGC-$3'$ (R). Primer sequences of U6: $5'$-GCTTCGGCACATATACTAAAAT-$3'$ (F) and $5'$-CGCTT CACGAATTTGCGTGTCAT-$3'$ (R).

### Target gene prediction

The miRWalk website (http://mirwalk.umm.uni-heidelberg.de/) was used to predict the target genes of miR-455-3p. Candidate genes were identified by setting the screening conditions (binding score as 1, acting position as 3' UTR and both in miRDB and miRTarBase).

### PPI network construction

Interaction analysis between miR-455-3p targets using the STRING website (https://cn.string-db.org/) and constructing a PPI network. Active interaction sources for PPI included textmining, experiments, databases, co-expression, neighbourhood, gene fusion and co-occurrence. The interaction score was set to 0.4 (medium confidence).

### KEGG pathway analysis

KEGG pathways were enriched by calculating strength (enrichment index) and false discovery rate (FDR) based on PPI network counts. The top 20 enriched pathways with FDR < 0.05 were displayed.

### Statistical analysis

SPSS 25.0 software was used for statistical analysis. The measurement data were expressed as t test and mean ± SD. The enumeration data were expressed as chi-square test. Adjust P values were obtained from multiple hypothesis test. P < 0.05 was considered statistically signiffcant.

## Results

### miR-455-3p is highly expressed in peripheral blood of PCa patients

To obtain DEmiRNAs in peripheral blood of PCa patients, we performed a comparative analysis between GSE206793 and GSE112264 datasets. For GSE206793 dataset, 46 DEmiRNAs were obtained by screening with a thresholds of |log2 (FC)| > 1 and P < 0.05 (Fig 1A). Due to the large samples in GSE112264 dataset, 604 DEmiRNAs were obtained by dimensionality reduction with a higher screening threshold of |log2 (FC)| > 3 and P < 0.01 (Fig 1B). We found that miR-455-3p co-occurred in both datasets (Fig 1C). Furthermore, miR-455-3p was highly expressed in the peripheral blood of PCa patients compared to healthy individuals (Fig 2A and 2C). Fig 1D demonstrates the miR-455-3p expression in each sample of the GSE112264 dataset. These results suggest that miR-455-3p may be a risk factor for PCa development.

### miR-455-3p has a unique clinical feature in PCa

Further insight into the correlation between miR-455-3p and clinical features of PCa. Patients in the GSE206793 dataset were divided into low-risk, medium-risk and high-rsk groups

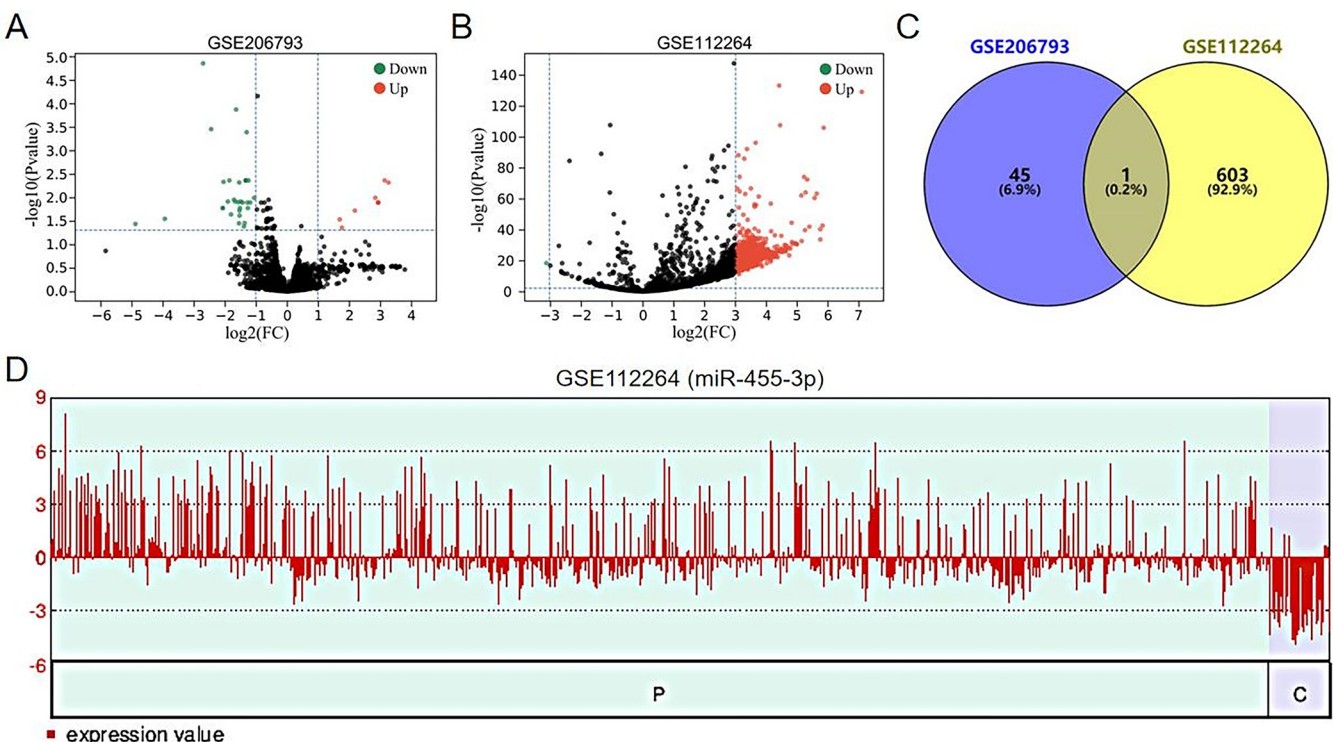

**Fig 1. Screening for common DEmiRNAs in datasets.** (A) Volcano map showing DEmiRNAs in GSE206793 dataset. (B) Volcano map showing DEmiRNAs in GSE112264 dataset. (C) Venn diagram showing common DEmiRNAs between GSE206793 and GSE112264 datasets. (D) miR-455-3p expression in each sample of GSE112264 dataset.

according to the EAU-ESTRO-SIOG Guidelines on Prostate Cancer. Patients in the GSE112264 dataset were divided into T1, T2, T3 and T4 groups according to T stage. We found that miR-455-3p expression was not associated with risk degree and T stage (Fig 2B and 2D). Then, patients in the GSE206793 dataset were divided into two groups based on age ($< 60$ vs $\geq 60$), GS ($< 8$ vs $\geq 8$) and PSA ($< 4$ vs $\geq 4$) for comparison. The results revealed that miR-455-3p was highly expressed in patients with GS $\geq 8$, independent of age and PSA (Fig 3A–3C). In addition, GSE206793 dataset also verified that there was no linear correlation between PSA level and miR-455-3p expression (r = 0.011, P = 0.226) (S1 Fig). The miR-455-3p expression in peripheral blood of other cancer patients was further observed. Compared with PCa, miR-455-3p was highly expressed in lung cancer, glioma and bladder cancer, suggesting a risk feature in cancer (Fig 2E). These results indicate that miR-455-3p is a risk factor for PCa.

## miR-455-3p has high diagnostic efficacy for PCa

To validate the diagnostic efficacy of miR-455-3p, ROC analysis was performed on both datasets. ROC is a coordinate diagram for statistical analysis formed with the false-positive rate on the horizontal axis and the true-positive rate on the vertical axis. Area under the curve (AUC) is the most important measure of ROC analysis. Typically, the AUC value ranges from 0.5 to 1.0. The higher the AUC value, the better the diagnostic efficacy. The results indicated that miR-455-3p had favorable diagnostic efficacy in GSE206793 and GSE112264 datasets, with AUC values of 0.943 and 0.847, respectively (Fig 3D and 3E). This indicates that miR-455-3p can be a novel diagnostic marker for PCa.

To clarify the diagnostic efficacy of miR-455-3p, we collected peripheral blood samples from 4 healthy individuals and 8 PCa patients for expression validation. qRT-PCR results

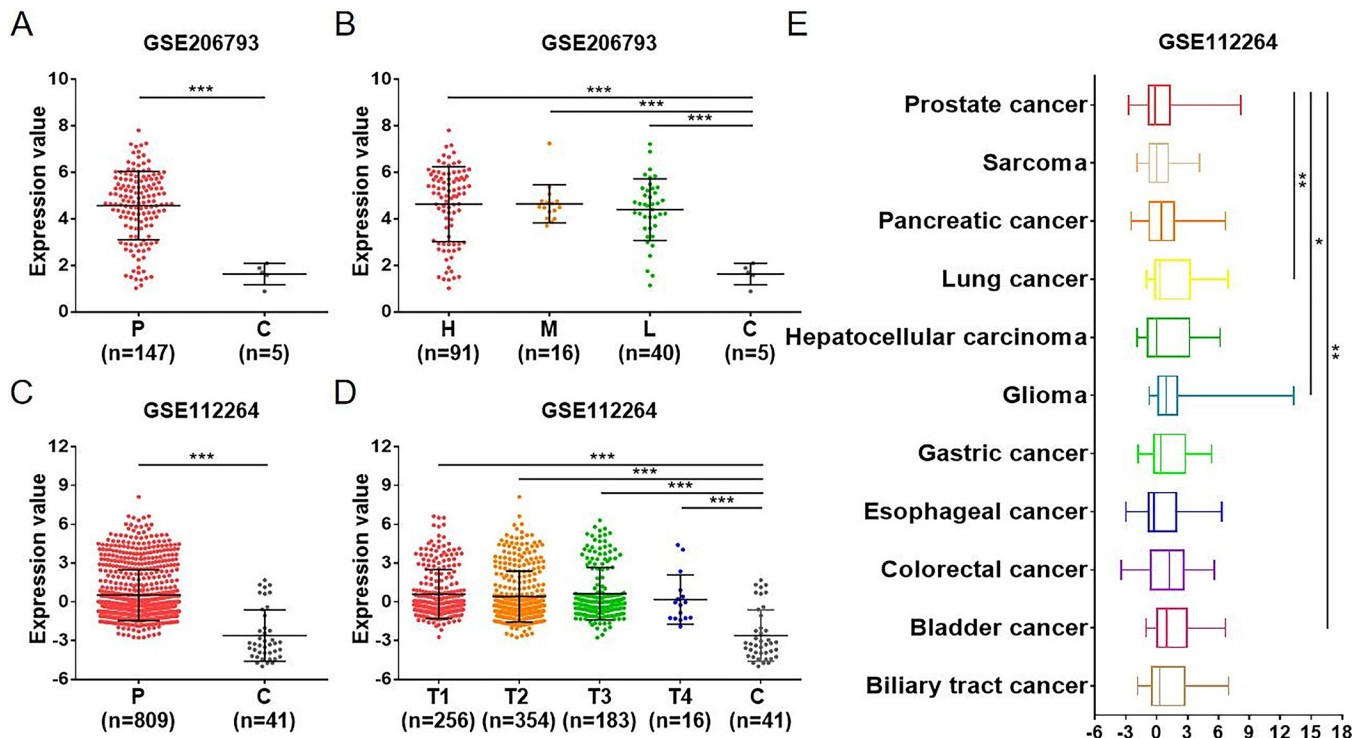

**Fig 2. Expression analysis of miR-455-3p in PCa and other cancers.** (A) Comparison of miR-455-3p expression between patient (P) and control (C) groups in GSE206793 dataset. (B) Comparison of miR-455-3p expression among low-risk (L), medium-risk (M), high-risk (H) and C groups in GSE206793 dataset. (C) Comparison of miR-455-3p expression between P and C groups in GSE112264 dataset. (D) Comparison of miR-455-3p expression among T1, T2, T3, T4 and C groups in GSE112264 dataset. (E) Comparison of miR-455-3p expression between PCa and 10 other cancers in GSE112264 dataset (*P < 0.05, **P < 0.01, ***P < 0.001).

demonstrated that miR-455-3p was highly expressed in patients with GS ≥ 8, which was consistent with the analyzed results of the GSE206793 dataset (Fig 4A and 4B). The diagnostic efficacy between miR-455-3p and PSA was further compared. The serum PSA levels of 8 PCa patients were obtained based on their case reports, and the comparison revealed that PSA levels were not associated with GS (Fig 4C). Interestingly, 2 patients (P1 and P6) had aberrant serum PSA levels of 3.38 and 4.45 ng/ml, while their GS was 9 (5 + 4) and 6 (3 + 3), respectively (Fig 4D). PSA was misdiagnosed in P1 and had low diagnostic efficacy in P6, while the corresponding miR-455-3p was highly expressed (Fig 4D). MRI films of P1 and P6 further demonstrated their lesions (Fig 4E). Conclusively, we speculate that miR-455-3p may possess a superior diagnostic efficacy to PSA. To further highlight the superiority of peripheral blood miR-455-3p as a diagnostic marker for PCa, we compared it with urinary cell-free miR-455-3p in GSE138740 dataset. We found that urinary cell-free miR-455-3p expression was not associated with PCa development and GS (S2A and S2B Fig). These results illustrate the strong diagnostic potential of peripheral blood miR-455-3p for PCa.

## miR-455-3p has superior diagnostic efficacy to PSA

To further verify our speculation, we obtained 20 PCa patients with serum PSA < 4 ng/ml from GSE206793 dataset for expression validation. Comparison among all PCa patients revealed that PSA levels were not associated with GS, which was consistent with the clinical samples detection (Fig 5A). Expression validation against 20 misdiagnosed patients showed that miR-455-3p was highly expressed in these patients compared to 5 healthy individuals

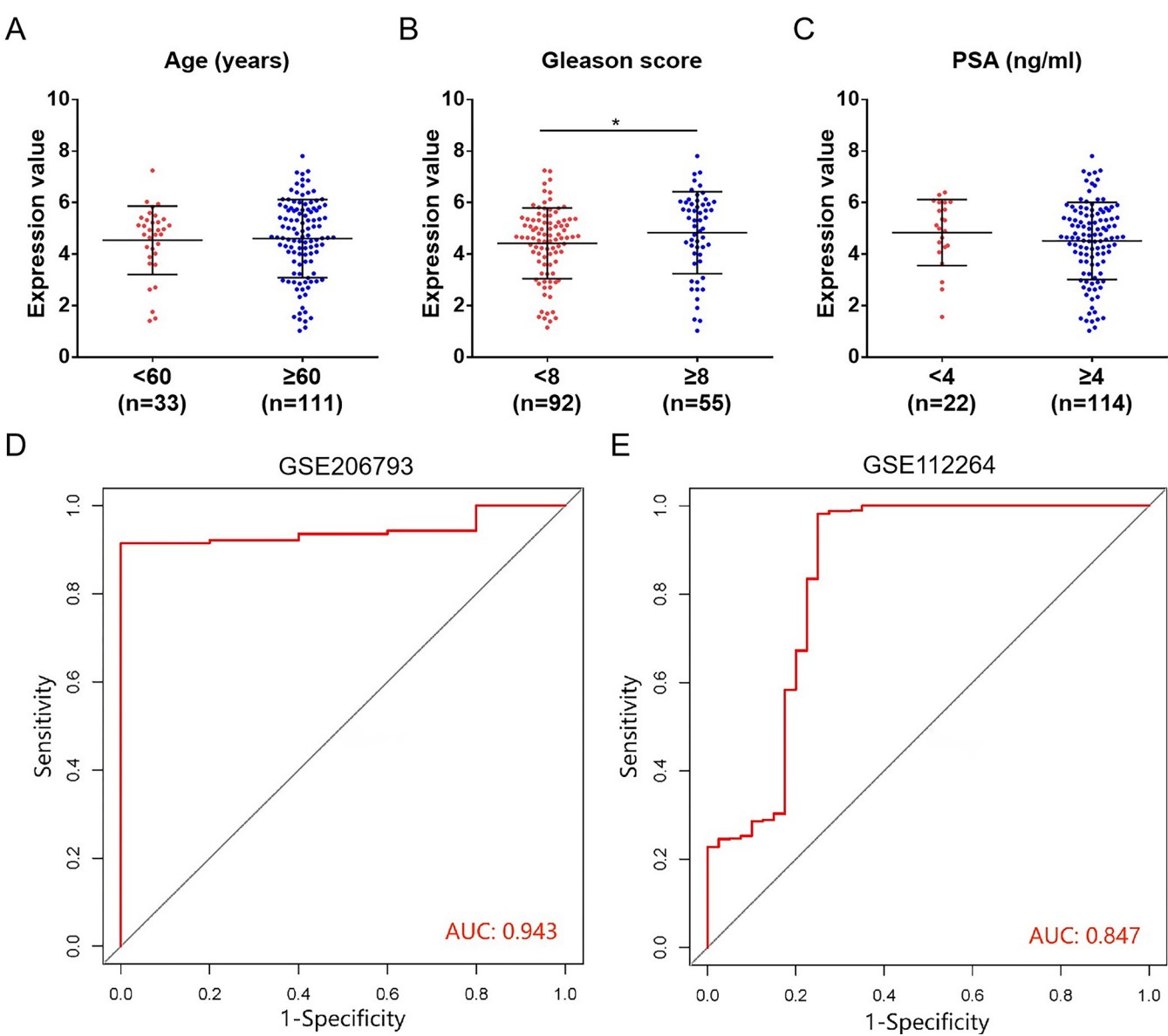

**Fig 3. Correlation analysis of miR-455-3p with clinical features in PCa.** (A) Comparison of miR-455-3p expression between patients aged $< 60$ and $\geq 60$ in GSE206793 dataset. (B) Comparison of miR-455-3p expression between patients with GS $< 8$ and $\geq 8$ in GSE206793 dataset. (C) Comparison of miR-455-3p expression between patients with PSA $< 4$ and $\geq 4$ in GSE206793 dataset. (D) Diagnostic efficacy of miR-455-3p in GSE206793 dataset by ROC evaluation. (E) Diagnostic efficacy of miR-455-3p in GSE112264 dataset by ROC evaluation (*P $< 0.05$).

(Fig 5B). In addition, miR-455-3p was highly expressed in patients with GS $\geq 8$ compared to those with GS $< 8$, further emphasizing its correlation with high risk of PCa (Fig 5C). Fig 5D demonstrated the serum PSA level with miR-455-3p expression in each misdiagnosed patient, which better illustrated the superior diagnostic efficacy of miR-455-3p to PSA.

## PPP2R2A, ITGB1 and CDKN1A are key targets of miR-455-3p for cancer regulation

Finally, we provide insight into the related molecular mechanisms of miR-455-3p involved in the regulation of biological functions. Based on the transcriptional repression effect of

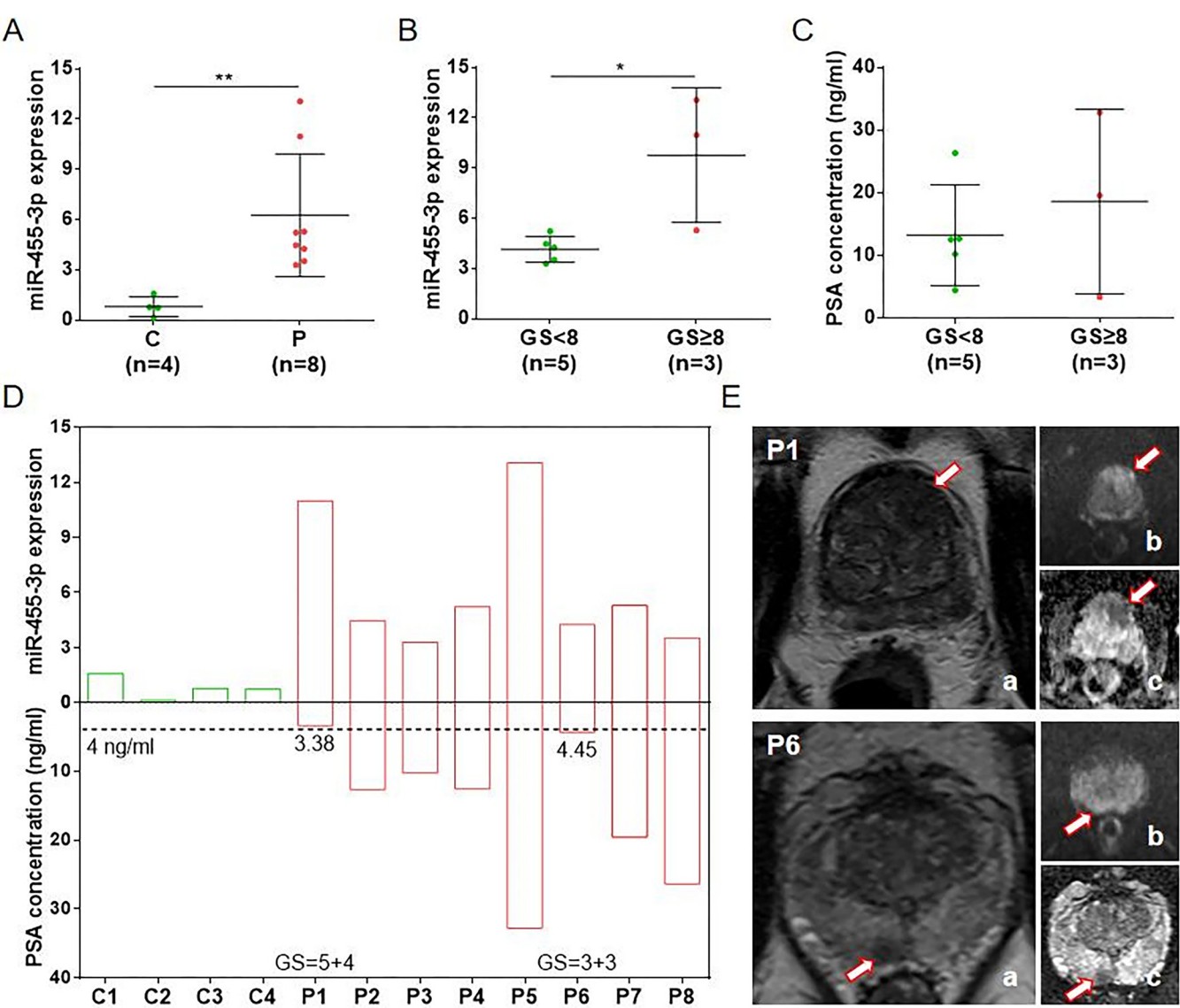

**Fig 4. Diagnostic efficacy validation of miR-455-3p in clinical samples.** (A) Comparison of miR-455-3p expression between clinical patients (P) and healthy individuals (C). (B) Comparison of miR-455-3p expression between patients with GS < 8 and ≥ 8. (C) Comparison of PSA concentration between patients with GS < 8 and ≥ 8. (D) Comparison of miR-455-3p expression and PSA concentration in each clinical sample. (E) Diagnostic films by MRI in patient 1 (P1) and patient 6 (P6); (a) focal low-signal shadow in the left lobe (P1) and in the peripheral band of the right lobe (P6) of the prostate gland; (b) slightly high signal on diffusion-weighted imaging (b = 1000); (c) apparent diffusion coefficient shows low signal shadow (*P < 0.05, **P < 0.01).

miRNAs, 9 potential target genes of miR-455-3p (PPP2R2A, ATL2, TOR1B, POFUT2, CDKN1A, GIGYF2, ITGB1, GGCX and OTULINL) were successfully predicted using miR-Walk website. The PPI network was further constructed for 9 target proteins using STRING website, in which PPP2R2A, ITGB1 and CDKN1A could be grouped into 3 clusters, respectively (Fig 6A). KEGG pathway analysis revealed that these proteins were enriched in various cancers including PCa (such as small cell lung cancer, bladder cancer, glioma, pancreatic cancer and melanoma), biological processes (such as cell cycle, cellular senescence and focal adhesion) and molecular signals (such as p53, PI3K/Akt and FoxO signaling pathways) (Fig 6B). These studies provide a basis for the molecular biology of miR-455-3p mediating PCa development.

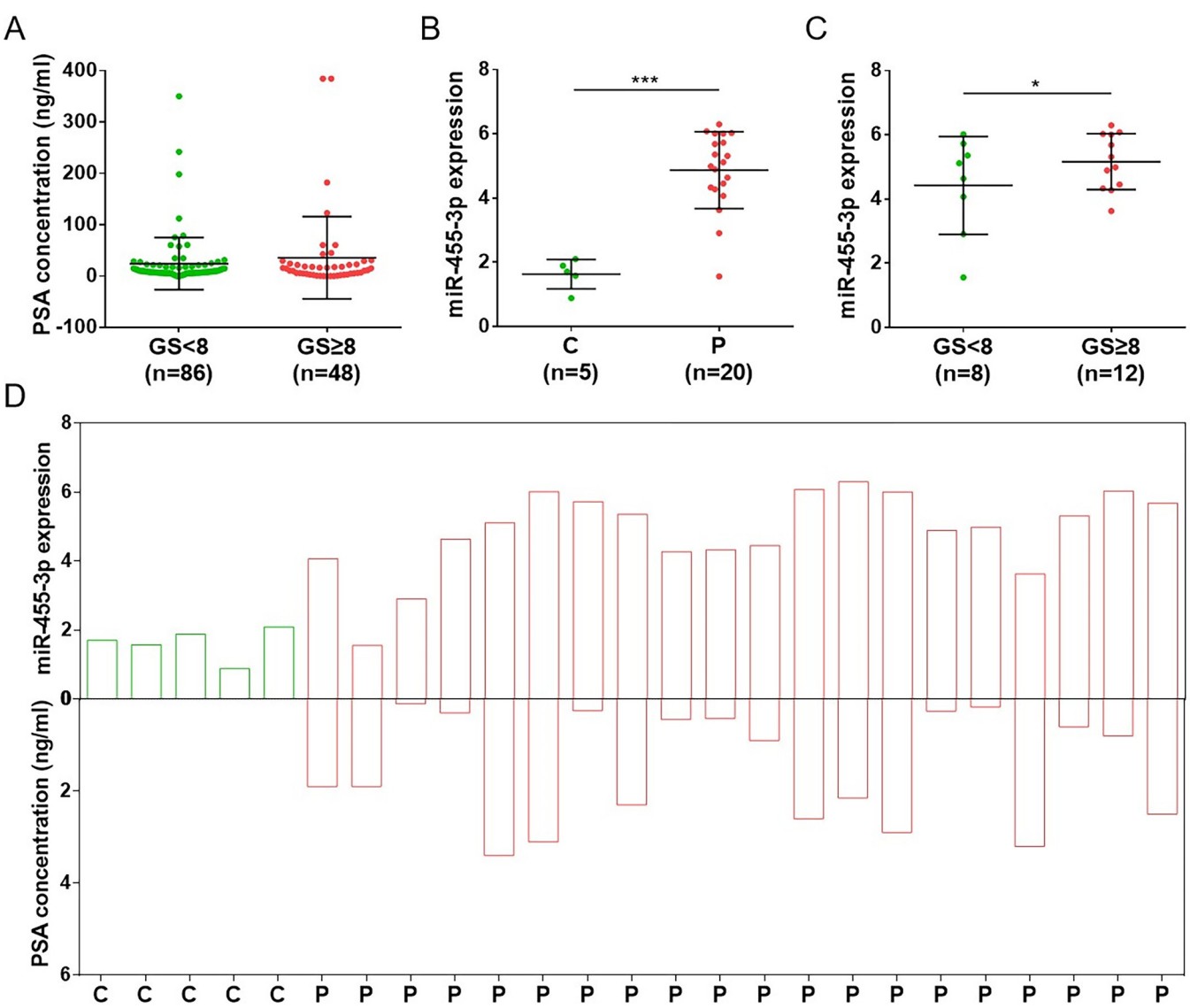

**Fig 5. Diagnostic efficacy validation of miR-455-3p in GSE206793 dataset.** (A) Comparison of PSA concentration between all patients with GS < 8 and ≥ 8. (B) Comparison of miR-455-3p expression between misdiagnosed patients (P) and healthy individuals (C). (C) Comparison of miR-455-3p expression between misdiagnosed patients with GS < 8 and ≥ 8. (D) Comparison of miR-455-3p expression and PSA concentration in each sample (*P < 0.05, ***P < 0.001).

## Discussion

PCa is a common clinical disease with a complex pathogenesis. Decreased immunity or changes in intracellular genetic material will lead to uncontrolled or abnormal cell growth [17–19]. Although there have been some studies on circulating miRNAs as diagnostic markers for PCa, their diagnostic roles are still controversial, such as inconsistent results from different data sources [20]. In addition, individual differences in patients and tumor heterogeneity can affect the diagnostic efficacy of circulating miRNAs, thus hardly forming a unified conclusion [21–24].

The present study was based on two datasets (GSE206793 and GSE112264) expressing miR-NAs in peripheral blood of PCa patients for diagnostic analysis. By setting reasonable screening thresholds, it was found that miR-455-3p co-occurred in both datasets and was highly

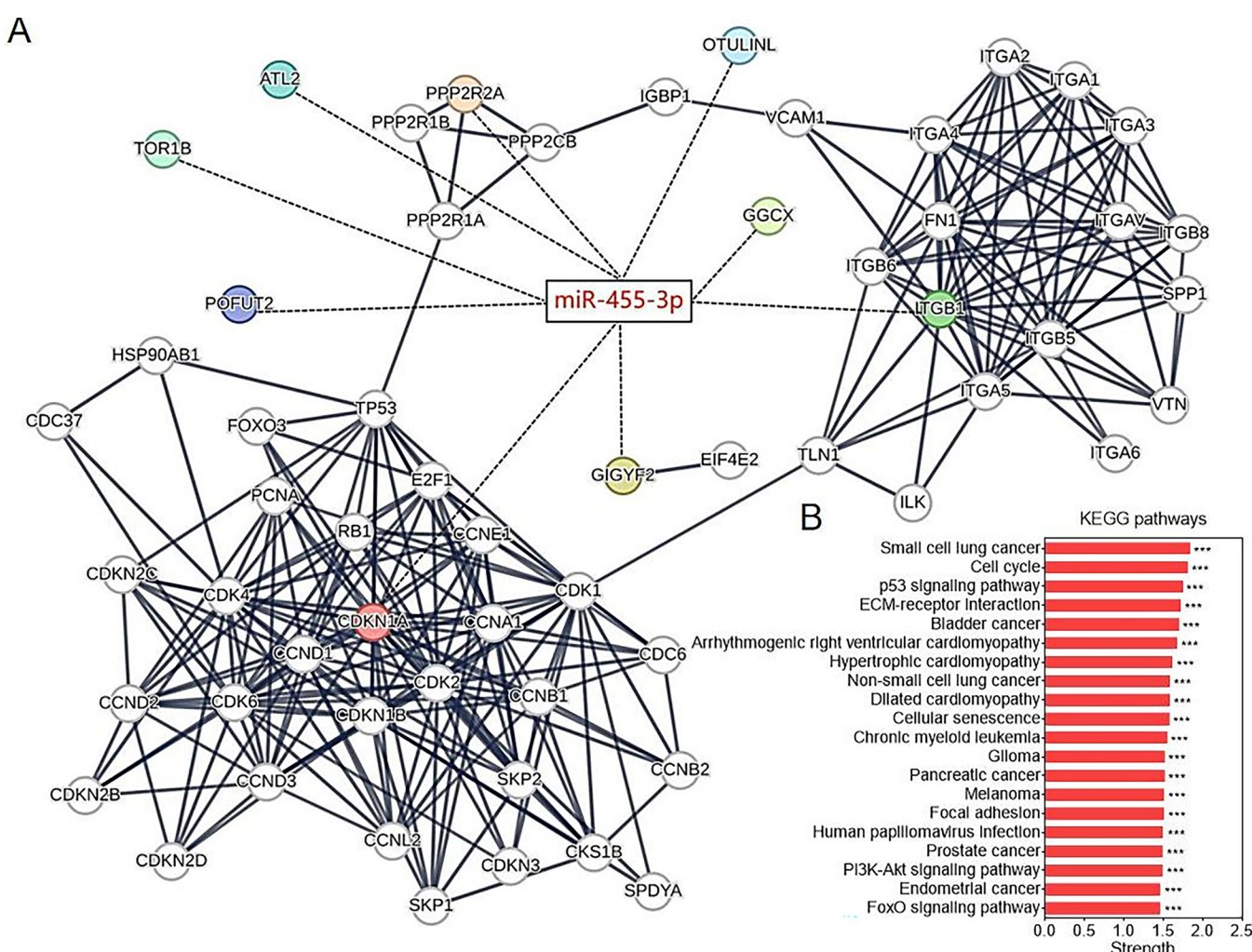

**Fig 6. PPI network construction and KEGG pathway analysis.** (A) Protein-protein interactions were analyzed for 9 potential targets of miR-455-3p using STRING website. (B) Enrichment analysis of these targets in cancer, biological process and molecular signal based on PPI network.

expressed in PCa patients. In addition, miR-455-3p was highly expressed in patients with high GS, independent of T stage, age and PSA. Serum miR-455-3p was also highly expressed in other cancer patients, especially lung cancer, glioma and bladder cancer. These findings revealed a unique clinical feature of miR-455-3p in PCa. In glioma, miR-455-3p is highly expressed in patient tissues and represents an independent prognostic indicator of overall survival in patients, promising to serve as a prognostic biomarker [25]. In liver cancer (LC), there was a 1.022-fold increase in the odds ratio for each unit increase in miR-455-3p, and increased serum levels of miR-122-5p and miR-455-3p were independently associated with an increased risk of incident LC in type 2 diabetes (T2D), which may be the potential biomarkers for early diagnosis of LC in T2D [26]. Moreover, miR-455-3p has a potential diagnostic role in other non-cancer diseases, such as acute graft-versus-host disease, Alzheimer's disease and acute myocardial infarction [27–29]. Consistent with these studies, ROC analysis demonstrated a favorable diagnostic efficacy of miR-455-3p in PCa patients (AUC > 0.8).

To validate the diagnostic efficacy of miR-455-3p, clinical blood samples were collected for assay. Consistent with the analysis of both datasets, miR-455-3p was highly expressed in PCa

patients with GS $\geq$ 8, while PSA levels were not associated with GS. Among the 8 PCa patients, P1 with high GS (5 + 4) had a misdiagnosis of serum PSA < 4 ng/ml, while P6 with medium GS (3 + 3) had a serum PSA of 4.45 ng/ml, which was inconsistent with the risk degree. Clinically, serum PSA $\geq$ 4 ng/ml suggests a risk of developing PCa [30]. However, PSA has the disadvantages of high false-positive rate, low sensitivity and inability to distinguish the malignancy degree of the tumor, failing to accurately diagnose and predict PCa development [31–33]. However, miR-455-3p was highly expressed in both P1 and P6, suggesting that it may have a superior diagnostic efficacy to PSA. Further validation using the GSE206793 dataset also revealed similar results. miR-455-3p was highly expressed in 20 misdiagnosed PCa patients (PSA < 4 ng/ml). Our study identified that miR-455-3p could be used as a novel diagnostic marker for PCa. Although miR-455-3p may have an advantage in diagnostic efficacy compared with PSA, a larger sample amount is still needed for validation. Furthermore, we unexpectedly found that miR-455-3p exhibited non-diagnostic efficacy in the urine of PCa patients, suggesting its specificity as a diagnostic marker in peripheral blood.

Finally, the downstream targets of miR-455-3p were predicted by miRWalk website and 9 potential genes (PPP2R2A, ATL2, TOR1B, POFUT2, CDKN1A, GIGYF2, ITGB1, GGCX and OTULINL) were successfully screened. PPI network revealed that PPP2R2A, ITGB1 and CDKN1A were the key nodes. KEGG results indicated that they were closely related to cancer and involved in the regulation of various biological processes and molecular signals. In PCa, upregulation of miR-556-5p promoted cell proliferation by suppressing PPP2R2A expression [34]. In bladder cancer, miR-222 attenuated cisplatin-induced cell death by targeting the PPP2R2A/Akt/mTOR axis [35]. In addition, ITGB1 and CDKN1A were found to play key regulatory roles in PCa [36–39]. These results further suggest an important mediating role of miR-455-3p in PCa development.

In conclusion, miR-455-3p is highly expressed in the peripheral blood of PCa patients and correlates with GS. miR-455-3p is a novel diagnostic marker for PCa with superior diagnostic potential to PSA.

## Supporting information

**S1 Fig. Correlation between PSA concentration and miR-455-3p expression in GSE206793 dataset.**
(TIF)

**S2 Fig. Correlation analysis of urinary cell-free miR-455-3p with clinical features in PCa.**
(A) Comparison of miR-455-3p expression between P and C groups in GSE138740 dataset. (B) Comparison of miR-455-3p expression between patients with GS < 8 and $\geq$ 8 in GSE138740 dataset.
(TIF)

**S1 Data.**
(XLSX)

## Author Contributions

**Conceptualization:** Yi Cen, Haiyan Wang, Guangbin Zhu.

**Data curation:** Yuyu Xu, Churuo Zhang, Xiangjin Lin, Xuan Ye.

**Formal analysis:** Churuo Zhang, Zeyu Zha.

**Methodology:** Yi Cen.

**Project administration:** Shourui Feng, Haiyan Wang.

**Resources:** Xiangjin Lin.

**Software:** Yuyu Xu, Zeyu Zha.

**Supervision:** Haiyan Wang, Guangbin Zhu.

**Visualization:** Yi Cen, Shourui Feng.

**Writing – original draft:** Yi Cen, Shourui Feng.

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
