## [Decision Letter · Decision Letter 0]

8 Nov 2024

PONE-D-24-12980miR-455-3p has superior diagnostic potential to PSA in peripheral blood for prostate cancerPLOS ONE

Dear Dr. Zhu,

Thank you for submitting your manuscript to PLOS ONE. After careful consideration, we feel that it has merit but does not fully meet PLOS ONE’s publication criteria as it currently stands. Therefore, we invite you to submit a revised version of the manuscript that addresses the points raised during the review process.

We look forward to receiving your revised manuscript.

Kind regards,

Gurudeeban Selvaraj, PhD

Academic Editor

PLOS ONE

**Journal Requirements:**

National Natural Science Foundation of China (Grant Number: 82201522)

4. We notice that your supplementary figures are included in the manuscript file. Please remove them and upload them with the file type 'Supporting Information'. Please ensure that each Supporting Information file has a legend listed in the manuscript after the references list.

Reviewers' comments:

Reviewer's Responses to Questions

**Comments to the Author**

1. Is the manuscript technically sound, and do the data support the conclusions?

Reviewer #1: Yes

2. Has the statistical analysis been performed appropriately and rigorously? 

Reviewer #1: Yes

3. Have the authors made all data underlying the findings in their manuscript fully available?

Reviewer #1: Yes

4. Is the manuscript presented in an intelligible fashion and written in standard English?

Reviewer #1: Yes

5. Review Comments to the Author

**Reviewer #1:** The manuscript evaluated miR-455-3p and PSA in the diagnosis of prostate cancer; it is well-designed; however, some flaws should be considered as follows:

Please determine the internal control and housekeeping gene in the material method.

Please use the reference for the Livak method (2-∆∆CT).

It would be better to check urine Cell-Free MicroRNAs in prostate cancer patients, and compare it with serum concentration.

Please explain the number of patients and healthy control in the material method, in sample collection the number of patients is not enough (PCa:8, healthy:4); however, in the result it was much more, so clarify the number of samples in the Clinical Sample Collection. Also, mention if qPCR was conducted on all the samples.

It would better to estimate the correlation (r) between the concentration of PSA and the relative gene expression value in patients and healthy control.

6. PLOS authors have the option to publish the peer review history of their article (what does this mean?). If published, this will include your full peer review and any attached files.

Reviewer #1: No

---

## [Author Response · Author response to Decision Letter 0]

18 Nov 2024

Please determine the internal control and housekeeping gene in the material method.

Thank you for your feedback. We have revised the manuscript accordingly. In the qPCR analysis, we have included U6 as the internal control gene to normalize the expression levels. This modification is now reflected in the materials and methods section.

Please use the reference for the Livak method (2-∆∆CT).

Thank you for your suggestion. We have now included the appropriate reference for the Livak method (2^-ΔΔCT) in the manuscript:

Schmittgen TD, Livak KJ. Analyzing real-time PCR data by the comparative C(T) method. Nat Protoc. 2008;3(6):1101-8. doi: 10.1038/nprot.2008.73. PMID: 18546601.

It would be better to check urine Cell-Free MicroRNAs in prostate cancer patients and compare it with serum concentration.

Thank you for the insightful suggestion. We have incorporated an analysis of urine cell-free miRNAs by adding GSE138740 dataset: Additionally, we included Figure S2, which presents correlation analyses of urinary cell-free miR-455-3p with clinical features in prostate cancer (PCa) patients. Specifically:

Figure S2A shows a comparison of miR-455-3p expression between prostate cancer (P) and control (C) groups in the GSE138740 dataset.

Figure S2B compares miR-455-3p expression between patients with Gleason scores (GS) < 8 and ≥ 8 in the GSE138740 dataset.

Please explain the number of patients and healthy control in the material method, in sample collection the number of patients is not enough (PCa:8, healthy:4); however, in the result it was much more, so clarify the number of samples in the Clinical Sample Collection. Also, mention if qPCR was conducted on all the samples.

Thank you for pointing this out. In the revised materials and methods section, we have clarified the number of samples used in our analysis:

The results presented in Figures 1–3 and 5 are based on analyses of public datasets (GSE206793 and GSE112264). Figure 4 results reflect analyses conducted on clinical samples, which included qPCR validation.

For the clinical sample collection, we used a total of 8 prostate cancer (PCa) samples and 4 healthy controls for qPCR analysis, as described. The qPCR was conducted on all clinical samples collected, and this has been clarified in the materials and methods section.

It would better to estimate the correlation (r) between the concentration of PSA and the relative gene expression value in patients and healthy control.

Thank you for the suggestion. We have conducted a correlation analysis between PSA concentration and miR-455-3p expression and presented the results in Figure S1, which shows the correlation in the GSE206793 dataset.

---

## [Decision Letter · Decision Letter 1]

27 Dec 2024

miR-455-3p has superior diagnostic potential to PSA in peripheral blood for prostate cancer

PONE-D-24-12980R1

Dear Dr. Zhu,

We’re pleased to inform you that your manuscript has been judged scientifically suitable for publication and will be formally accepted for publication once it meets all outstanding technical requirements.

Kind regards,

Gurudeeban Selvaraj, PhD

Academic Editor

PLOS ONE

Reviewers' comments:

Reviewer's Responses to Questions

**Comments to the Author**

1. If the authors have adequately addressed your comments raised in a previous round of review and you feel that this manuscript is now acceptable for publication, you may indicate that here to bypass the “Comments to the Author” section, enter your conflict of interest statement in the “Confidential to Editor” section, and submit your "Accept" recommendation.

Reviewer #1: All comments have been addressed

Reviewer #2: All comments have been addressed

2. Is the manuscript technically sound, and do the data support the conclusions?

Reviewer #1: Yes

Reviewer #2: Yes

3. Has the statistical analysis been performed appropriately and rigorously? 

Reviewer #1: Yes

Reviewer #2: Yes

4. Have the authors made all data underlying the findings in their manuscript fully available?

Reviewer #1: Yes

Reviewer #2: Yes

5. Is the manuscript presented in an intelligible fashion and written in standard English?

Reviewer #1: Yes

Reviewer #2: Yes

6. Review Comments to the Author

Reviewer #1: (No Response)

Reviewer #2: The detection of PSA, a commonly utilized tumor marker for prostate cancer (PCa), is often influenced by medications and various non-cancerous prostate lesions, highlighting the need for a novel marker to enhance diagnostic specificity and sensitivity. This study, leveraging bioinformatics techniques, has successfully identified miR-455-3p as a promising new marker. The article offers valuable insights into the advantages of utilizing miR-455-3p as an alternative PCa marker and delves into the intricate molecular mechanisms underlying its biological functions. The authors have thoughtfully addressed and revised the comments provided by previous reviewers. The overall structure and logic of the article are clear and coherent, making it a potential new tool for the precise diagnosis of clinical PCa. Therefore, I warmly recommend the acceptance of this manuscript.

7. PLOS authors have the option to publish the peer review history of their article (what does this mean?). If published, this will include your full peer review and any attached files.

Reviewer #1: **Yes: **Elham Kazemirad

Reviewer #2: No

---

## [Editor Report · Acceptance letter]

5 Jan 2025

PONE-D-24-12980R1 

PLOS ONE

Dear Dr. Zhu, 

I'm pleased to inform you that your manuscript has been deemed suitable for publication in PLOS ONE. Congratulations! Your manuscript is now being handed over to our production team.

Kind regards, 

on behalf of

Dr. Gurudeeban Selvaraj, PhD 

Academic Editor

PLOS ONE